# DECENTRALIZED DECOUPLED TRAINING FOR FEDERATED LONG-TAILED LEARNING

## ABSTRACT

In the real world, the data samples often follow a long-tailed distribution, which poses a great challenge for Federated Learning (FL). That is, when the data is decentralized and long-tailed, FL may produce a poorly-behaved global model that is severely biased towards the head classes with the majority of the training samples. To settle this issue, decoupled training has recently been introduced to FL. Decoupled training aims to re-balance the biased classifier after the normal instance-balanced training, and has achieved promising results in centralized long-tailed learning. The current study directly adopts the decoupled training idea on the server side by re-training the classifier on a set of pseudo features, due to the unavailability of a global balanced dataset in FL. Unfortunately, this practice restricts the capacity of decoupled training in federated long-tailed learning as the low-quality pseudo features lead to a sub-optimal classifier. In this work, motivated by the distributed characteristic of FL, we propose a decentralized decoupled training mechanism by leveraging the abundant real data stored in the local. Specifically, we integrate the local real data with the global gradient prototypes to form the local balanced datasets, and thus re-balance the classifier during the local training. Furthermore, we introduce a supplementary classifier in the training phase to help model the global data distribution, which addresses the problem of contradictory optimization goals caused by performing classifier re-balancing locally. Extensive experiments show that our method consistently outperforms the existing state-of-the-art methods in various settings. Our code will be released upon acceptance.

## 1 INTRODUCTION

Federated Learning (FL) (McMahan et al., 2017) is proposed as an effective distributed learning framework to enable local clients to train a global model collaboratively without exposing their local private data to each other. However, the performance of FL is heavily hindered by the long-tailed/class-imbalanced data distribution phenomenon in the real world. That is, the global data distribution (i.e., the data distribution of the training samples merged from all clients' local data) usually shows a long-tailed pattern (Zhang et al., 2021; Wang et al., 2021), where head classes occupy a much larger proportion of the training samples than tail classes. Applying FL directly on such long-tailed data will produce a poorly performing global model that is severely biased to the head classes (Wang et al., 2021).

Dealing with FL on the non-i.i.d. and long-tailed data is challenging in two aspects: **First**, as the data samples are not identically and independently distributed (non-i.i.d.) across different clients in FL (McMahan et al., 2017), the local data distributions (i.e., local imbalance) show inconsistent long-tailed patterns compared to that of the global data distribution (i.e., global imbalance) (Wang et al., 2021). Thus, tackling the local imbalance problem only (e.g., Fed-Focal Loss (Sarkar et al., 2020)) will not help to address the global imbalance problem in FL. **Second**, considering the data privacy concern, it is infeasible to explicitly obtain the imbalance pattern of the global data distribution from the local data information. This further limits the application of the global class-level re-weighting strategy (Cui et al., 2019).[1]

Decoupled training is first proposed in centralized long-tailed learning (Kang et al., 2019; Zhou et al., 2020), which disentangles the long-tailed learning into the representation learning phase

---

[1]Also, class-level re-weighting practice itself is an improper solution according to Kang et al. (2019).

and the classifier learning phase. These works (Kang et al., 2019; Zhou et al., 2020) find that the instance-balanced training (i.e., uniform sampling on the entire training set to make the contribution of each sample the same) leads to well-learned representations but a biased classifier. Thus, they propose to re-train the classifier on a small balanced dataset after the instance-balanced training and have achieved promising results. We believe the decoupled training idea is also suitable for tackling the class-imbalanced problem in FL, as it can adjust the biased classifier at the global level without the necessity of obtaining the global imbalance pattern.

Nevertheless, applying decoupled training into FL faces a great challenge that there lacks a public balanced dataset for re-training classifier due to the data privacy concern. A recent study CReFF (Shang et al., 2022b) directly performs decoupled training on the server side, by re-training the classifier on a set of pseudo features created on the server. However, we argue that adjusting the classifier in such a centralized manner neglects the decentralized characteristic of FL, and only produces a sub-optimal classifier caused by the poor-quality pseudo features with high similarity per class.

In this paper, we propose the **decentralized decoupled training** mechanism to realize the full potential of decoupled training in federated long-tailed learning. Our main idea is to fully utilize the abundant real data that is only stored in the clients. Therefore, we are motivated to allow clients to re-balance the classifier during local training. Specifically, we make each client re-balance the classifier on a local balanced dataset that is mixed with the local real data and the global gradient prototypes of the classifier sent by the server, while the latter is supposed to address the issue of missing classes in the local datasets. In this case, the classifier re-balancing is performed on the client side and in a decentralized way. Moreover, we add a supplementary classifier in the training phase to jointly model the global data distribution. This practice helps to overcome the optimization problem brought by the practice of local classifier re-balancing. Compared with CReFF, our fully-decentralized paradigm allows the clients to collaboratively train a balanced classifier with their sufficient local real data during local training, which needs no extra requirements on the server and produces an optimal classifier with better generalization ability. We conduct extensive experiments on three popular long-tailed image classification tasks, and the results show that our method can significantly outperform all existing federated long-tailed learning methods in various settings.

## 2 RELATED WORK

### 2.1 FEDERATED LEARNING

Federated Averaging (FedAvg) (McMahan et al., 2017) is the most widely-used FL algorithm, but it has been shown that the performance of FedAvg drops greatly when the data is non-i.i.d. (Karimireddy et al., 2020b). Therefore, plenty of existing FL studies (Li et al., 2018; Acar et al., 2020; Karimireddy et al., 2020b; Hsu et al., 2019; Reddi et al., 2020; Karimireddy et al., 2020a) target on dealing with the non-i.i.d. data partitions in FL. However, these studies neglect another realistic and important data distribution phenomenon that the data samples usually show a long-tailed pattern.

### 2.2 LONG-TAILED/IMBALANCED LEARNING

In the real world, the data points usually show a long-tailed distribution pattern. Therefore, learning good models on the long-tailed/class-imbalanced data has been widely studied (Zhang et al., 2021) in the traditional centralized learning, and attracts more and more attention in the FL setting.

**Centralized Long-Tailed Learning** The methods in the centralized long-tailed learning can be mainly divided into three categories: (1) **Class-level re-balancing methods** that includes over-sampling training samples from tail classes (Chawla et al., 2002), under-sampling data points from head classes (Liu et al., 2008), or re-weighting the loss values or the gradients of different training samples based on the label frequencies (Cui et al., 2019; Cao et al., 2019) or the predicted probabilities of the model (Lin et al., 2017). (2) **Augmentation-based methods** aim to create more data samples for tail classes either from the perspective of the feature space (Chu et al., 2020; Zang et al., 2021) or the sample space (Chou et al., 2020). (3) **Classifier re-balancing mechanisms** are based on the finding that the uniform sampling on the whole dataset during training benefits the representation learning but leads to a biased classifier, so they design specific algorithms to adjust the classifier during or after the representation learning phase (Zhou et al., 2020; Kang et al., 2019).

**Federated Long-Tailed Learning** Recently, a few studies begin to focus on the class imbalance problem in FL. Fed-Focal Loss (Sarkar et al., 2020) directly applies Focal Loss (Lin et al., 2017) in the clients' local training, but it neglects the fact that the local imbalance pattern is inconsistent with the global imbalance pattern. Ratio Loss (Wang et al., 2021) utilizes an auxiliary dataset on the server (which is usually impractical in real cases) to estimate the global data distribution, and send the estimated information to clients to perform class-level re-weighting during local training. CLIMB (Shen et al., 2021) is proposed as a client-level re-weighting method to give more aggregation weights to the clients with larger local training losses. However, both Ratio Loss and CLIMB bring negative effects to the representation learning due to the re-weighting practice (Kang et al., 2019), thus the improvement brought by them is limited. FEDIC (Shang et al., 2022a) also needs the impractical assumption to own an auxiliary balanced dataset and amount of unlabeled data for fine-tuning and performing knowledge distillation on the global model on the server. Most recently, CReFF (Shang et al., 2022b) adopts the decoupled training idea to re-train the classifier on the server by creating a number of federated features for each class, and achieves previously state-of-the-art performance. However, the low quality and the limited quantity of federated features restrict its potential.

## 3 METHODOLOGY

### 3.1 PROBLEM DEFINITION

In FL, each client $k$ ($k = 1, \cdots, N$) has its own local dataset $\mathcal{D}_k$, and all clients form a federation to jointly train a good global model. Then, the optimization goal of FL can be formulated as

$$\boldsymbol{\theta}^* = \arg\min_{\boldsymbol{\theta}} L(\boldsymbol{\theta}) = \arg\min_{\boldsymbol{\theta}} \sum_{k=1}^{N} \frac{|\mathcal{D}_k|}{\sum_{i=1}^{N} |\mathcal{D}_i|} L(\boldsymbol{\theta}; \mathcal{D}_k), \tag{1}$$

where $|\mathcal{D}_k|$ represents the sample quantity of $\mathcal{D}_k$, $L(\cdot; \mathcal{D}_k)$ is the local training objective in client $k$.

Federated Averaging (FedAvg) (McMahan et al., 2017) is the most popular FL framework to solve the above optimization problem. Specifically, in each round, the server sends the latest global model $\boldsymbol{\theta}^{t-1}$ to all sampled clients $k \in \mathcal{C}^t$, and each client $k$ performs multiple updates on $\boldsymbol{\theta}^{t-1}$ with its local dataset $\mathcal{D}_k$ and gets the new model $\boldsymbol{\theta}_k^t$. Then it only sends the accumulated gradients $\boldsymbol{g}_k^t = \boldsymbol{\theta}^{t-1} - \boldsymbol{\theta}_k^t$ back to the server, which aggregates the collected gradients and updates the global model as:

$$\boldsymbol{\theta}^t = \boldsymbol{\theta}^{t-1} - \eta_s \frac{1}{|\mathcal{C}^t|} \sum_{k \in \mathcal{C}^t} \frac{|\mathcal{D}_k|}{\sum_{i \in \mathcal{C}^t} |\mathcal{D}_i|} \boldsymbol{g}_k^t, \tag{2}$$

where $\eta_s$ is the server learning rate, $|\mathcal{C}^t|$ is the number of clients participating in the current round.

**In this paper, we study the optimization problem of FL in which the global data distribution $\mathcal{D} = \bigcup_{\mathbf{k}} \mathcal{D}_{\mathbf{k}}$ is long-tailed.** Previous studies in centralized long-tailed learning (Kang et al., 2019) propose to decouple the training on the long-tailed classification tasks into representation learning and classifier learning phases, and point out that performing class-level re-weighting rather than instance-balanced training brings negative impact on the representation learning, and the imbalanced data distribution mainly affects the classifier learning. Then they achieve significant improvement by re-balancing the classifier on a balanced dataset after training. This finding is also verified in federated long-tailed learning (Shang et al., 2022b). Thus, **our main idea is to effectively re-balance the classifier when dealing with the long-tailed global data in FL**. However, different from the centralized training, there is a lack of the global balanced dataset in FL for re-balancing classifier. **Then, being aware of the data-decentralized property of FL, we are motivated to make clients re-balance the classifier locally during training by taking great advantage of their abundant local real data**.

### 3.2 OPTIMIZATION TARGET

We split the original model architecture/parameters $\boldsymbol{\theta} = (\boldsymbol{P}, \boldsymbol{W})$ into two parts: the representation encoder $\boldsymbol{P}$ and the classifier $\boldsymbol{W}$, and aim to re-balance $\boldsymbol{W}$ during the local training to make it behave well on the class-balanced data distribution $\mathcal{D}^{bal}$. However, re-balancing the classifier during (instead of after) the representation learning phase leads to a contradictory optimization target $\mathcal{T}_{con}$:

$$\mathcal{T}_{con} = \left\{ \min_{\boldsymbol{P}, \boldsymbol{W}} L(\boldsymbol{P}, \boldsymbol{W}; \bigcup_k \mathcal{D}_k), \quad \text{s.t.} \quad L(\boldsymbol{W}; \boldsymbol{P}, \mathcal{D}^{bal}) = \min_{\boldsymbol{W}_{\boldsymbol{P}}} L(\boldsymbol{W}_{\boldsymbol{P}}; \boldsymbol{P}, \mathcal{D}^{bal}) \right\}, \tag{3}$$

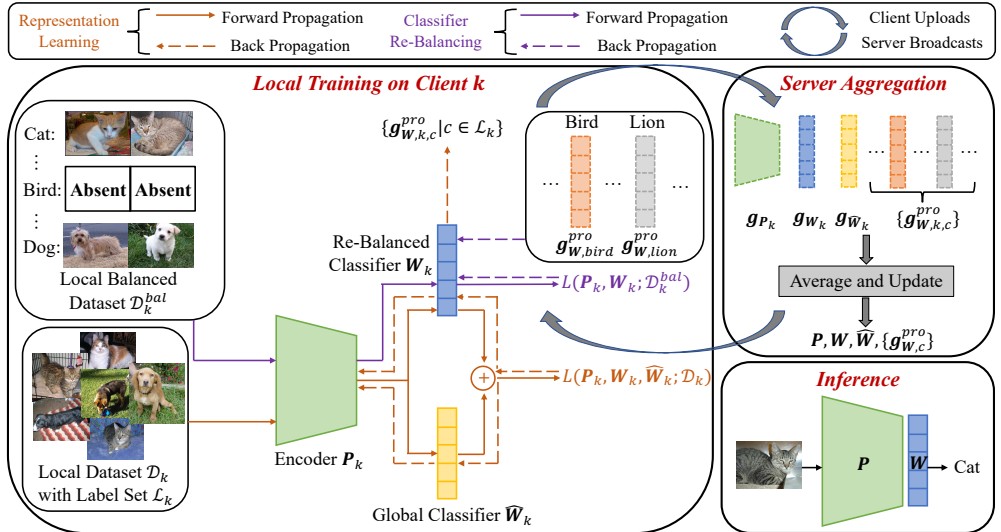

Figure 1: Full illustration of our method. We add a new global classifier $\widehat{W}$, and perform instance-balanced training on both the encoder and two classifiers. Additionally, we re-balance the original classifier during local training on a balanced dataset mixed with local real data and global gradient prototypes $\{g_{W,c}^{pro}\}$. In inference, we only keep the encoder $P$ and the re-balanced classifier $W$.

**where the first component before semi-colon in $\mathbf{L}(\cdot ; \cdot)$ is the variable while the latter is the constant condition.** The first part of Eq. (3) represents that based on FedAvg, the whole model is trained under the global data distribution. As for the second part, if we want to further re-balance the classifier $W$, the re-balanced classifier should also behave well on the balanced data distribution given the encoder is fixed. **However, when the global data distribution $\mathcal{D} = \bigcup_k \mathcal{D}_k$ is long-tailed, the above problem faces severe optimization difficulty, as one classifier can not be trained well to fit two different data distributions.**[2] To address the negative impact of performing local classifier re-balancing, we design an architecture of *the two-stream classifiers* by adding a supplementary classifier $\widehat{W}$ in the training phase (refer to Figure 1), in order to help model the global data distribution $\mathcal{D}$ and make re-balancing $W$ feasible.[3] We re-formulate our global optimization target as:

$$\mathcal{T} = \left\{ \min_{P, W, \widehat{W}} L(P, W, \widehat{W}; \bigcup_k \mathcal{D}_k), \quad \text{s.t.} \quad L(W; P, \mathcal{D}^{bal}) = \min_{W_P} L(W_P; P, \mathcal{D}^{bal}) \right\}. \quad (4)$$

**By making the combination of two classifiers model the global data distribution in the first part of Eq. (4), the problem of contradictory solutions of W is addressed.** Furthermore, it makes sure that the representation encoder is trained under the instance-balanced training paradigm, which benefits the representation learning most. In the following, we introduce our proposed algorithm to solve Eq. (4) from three aspects, including the local training, the server aggregation, and the inference stages. The full illustrations of our model architecture and training process are displayed in Figure 1.

### 3.3 CLASSIFIER RE-BALANCING BY INTEGRATING LOCAL REAL DATA WITH GLOBAL GRADIENT PROTOTYPES

#### 3.3.1 LOCAL TRAINING STAGE

In the local training, according to the basic idea of FedAvg framework, each client aims to solve the sub-problem of Eq. (4) as:

$$\mathcal{T}_k = \left\{ \min_{P, W, \widehat{W}} L(P, W, \widehat{W}; \mathcal{D}_k), \quad \text{s.t.} \quad L(W; P, \mathcal{D}^{bal}) = \min_{W_P} L(W_P; P, \mathcal{D}^{bal}) \right\}. \quad (5)$$

---

[2]We also consider another alternative two-stage classifier re-balancing paradigm in Appendix G, but the results show that it is much ineffective than the following introduced joint optimization paradigm.

[3]We explore the necessity of introducing this supplementary classifier in Section 5.2.

---

**Algorithm 1** Local Training Process of RedGrape

---

**Input**: Round number $t$, local data $\mathcal{D}_k$ with local label set $\mathcal{L}_k$, local model $(\boldsymbol{P}_k, \boldsymbol{W}_k, \widehat{\boldsymbol{W}}_k)$ initialized as received global model $(\boldsymbol{P}^{t-1}, \boldsymbol{W}^{t-1}, \widehat{\boldsymbol{W}}^{t-1})$, global gradient prototypes $\{\boldsymbol{g}_{\boldsymbol{W}^{t-2},c}^{pro}|c \in \mathcal{L}\}$.

1: Calculate local gradient prototypes $\{\boldsymbol{g}_{\boldsymbol{W}^t,k,c}^{pro}|c \in \mathcal{L}_k\}$ based on Eq. (11).
2: **for** Local step $i = 1, 2, \cdots, I$ **do**
3:     Update $\boldsymbol{P}$ and $\widehat{\boldsymbol{W}}$ based on Eq. (7).
4:     Compute batch gradients for $\boldsymbol{W}$ based on Eq. (8).
5:     Compute re-balancing gradients for $\boldsymbol{W}$ based on Eq. (9), Eq. (10) and Eq. (12).
6:     Update $\boldsymbol{W}$ based on Eq. (13).
7: **end for**
8: **return** Local gradients $(\boldsymbol{P}_k^t - \boldsymbol{P}^{t-1}, \boldsymbol{W}_k^t - \boldsymbol{W}^{t-1}, \widehat{\boldsymbol{W}}_k^t - \widehat{\boldsymbol{W}}^{t-1})$, local gradient prototypes $\{\boldsymbol{g}_{\boldsymbol{W}^{t-1},k,c}^{pro}|c \in \mathcal{L}_k\}$.

---

It is a constrained optimization problem that is non-trivial to solve, therefore, we manage to address it by applying the method of Lagrange Multipliers. That is, we turn it into the following unconstrained optimization problem by adding a penalty term on solving $\boldsymbol{W}$:

$$
\begin{aligned}
\mathcal{T}_k^{'} &= \min_{\boldsymbol{P}, \boldsymbol{W}, \widehat{\boldsymbol{W}}} L_k(\boldsymbol{P}, \boldsymbol{W}, \widehat{\boldsymbol{W}}) \\
&= \min_{\boldsymbol{P}, \boldsymbol{W}, \widehat{\boldsymbol{W}}} \left\{ L(\boldsymbol{P}, \boldsymbol{W}, \widehat{\boldsymbol{W}}; \mathcal{D}_k) + \lambda \left( L(\boldsymbol{W}; \boldsymbol{P}, \mathcal{D}^{bal}) - \min_{\boldsymbol{W_P}} L(\boldsymbol{W_P}; \boldsymbol{P}, \mathcal{D}^{bal}) \right) \right\}.
\end{aligned}
\tag{6}
$$

After choosing a proper $\lambda$, all parameters can be optimized by taking the derivative of $L_k$ over them.

**An overall look.** Before introducing technique details, we first give an overall look of the whole process of local training in our method in Algorithm 1. To be brief, the encoder parameters $\boldsymbol{P}$ and the supplementary classifier $\widehat{\boldsymbol{W}}$ will be trained under an instance-balanced manner (Line 3). When updating the original classifier $\boldsymbol{W}$, besides the gradients of the batch samples from $\mathcal{D}_k$ (Line 4), our method creates a local balanced dataset $\mathcal{D}_k^{bal}$ to help re-balancing $\boldsymbol{W}$ (Line 5-6) following the second target of Eq. (6). The detailed steps include the following parts:

**Updating $\boldsymbol{P}$ and $\widehat{\boldsymbol{W}}$, and calculating local batch gradients for $\boldsymbol{W}$.** In the $t$-th round, we perform the batch stochastic gradient decent mechanism[4] to update $\boldsymbol{P}$ and $\widehat{\boldsymbol{W}}$.[5] That is, for the local step $i = 1, 2, \cdots, I$, a random batch of examples $\mathcal{B}_k^i$ is sampled from $\mathcal{D}_k$ to perform that:

$$
\begin{aligned}
\boldsymbol{P}_k^i &= \boldsymbol{P}_k^{i-1} - \eta_l \nabla_{\boldsymbol{P}_k^{i-1}} L(\boldsymbol{P}_k^{i-1}, \boldsymbol{W}_k^{i-1}, \widehat{\boldsymbol{W}}_k^{i-1}; \mathcal{B}_k^i), \\
\widehat{\boldsymbol{W}}_k^i &= \widehat{\boldsymbol{W}}_k^{i-1} - \eta_l \nabla_{\widehat{\boldsymbol{W}}_k^{i-1}} L(\boldsymbol{P}_k^{i-1}, \boldsymbol{W}_k^{i-1}, \widehat{\boldsymbol{W}}_k^{i-1}; \mathcal{B}_k^i),
\end{aligned}
\tag{7}
$$

$\eta_l$ is the local learning rate. One important thing is, when calculating the above loss on each sample $(\boldsymbol{x}, y)$, the representation vector $\boldsymbol{h} := f(\boldsymbol{x}; \boldsymbol{P})$ will be first fed into both two classifiers and get two logits $\boldsymbol{W}^T \boldsymbol{h}$ and $\widehat{\boldsymbol{W}}^T \boldsymbol{h}$. Then we perform the element-wise addition to get the final logits $\boldsymbol{z} = \boldsymbol{W}^T \boldsymbol{h} + \widehat{\boldsymbol{W}}^T \boldsymbol{h}$, and use $\boldsymbol{z}$ for the loss calculation. Also, in the same forward and backward propagations, we can obtain the first part of gradients for $\boldsymbol{W}$ in Eq. (6) as

$$
\boldsymbol{g}_{\boldsymbol{W}_k^{i-1}}^{local} = \nabla_{\boldsymbol{W}_k^{i-1}} L(\boldsymbol{P}_k^{i-1}, \boldsymbol{W}_k^{i-1}, \widehat{\boldsymbol{W}}_k^{i-1}; \mathcal{B}_k^i).
\tag{8}
$$

**Calculating re-balancing gradients for $\boldsymbol{W}$.** For the second part of Eq. (6), it needs to calculate the gradients of $\boldsymbol{W}_k^{i-1}$ on a small balanced set $\mathcal{D}_k^{bal}$, which is supposed to be created in the local. However, there exists difficulty in constructing $\mathcal{D}_k^{bal}$ from $\mathcal{D}_k$, since **it is very likely that there are some classes missing in the local label set $\mathcal{L}_k$ of $\mathcal{D}_k$ due to the non-i.i.d. data partitions**. Thus, we propose a *mixed gradient re-balancing mechanism* to overcome this challenge by integ**R**ating local

---

[4]Here, we take SGD as an example, but we do not have the assumption on the type of local optimizer.
[5]For simplicity, we omit the bias term here, while our method is still applicable when the bias term exists.

rEal Data with Global gRAdient prototyPEs (**RedGrape** as our method). Specifically, for each class $c$, **(1)** if the sample quantity of class $c$ in $\mathcal{D}_k$ reaches a threshold $T$, we think client $k$ has sufficient samples of class $c$ in its local dataset, and randomly samples $T$ training samples of class $c$ (which can be different across rounds) to form $\mathcal{D}_{k,c}^{bal}$ for $\mathcal{D}_k^{bal}$. Then, the gradients contributed by class $c$ are

$$g_{\boldsymbol{W}_k^{i-1},c}^{bal} = \nabla_{\boldsymbol{W}_k^{i-1}} L(\boldsymbol{W}_k^{i-1}; \boldsymbol{P}_k^{i-1}, D_{k,c}^{bal}). \tag{9}$$

**(2)** If client $k$ does not have enough samples in class $c$, we choose to estimate the gradient contribution of class $c$ with the *global gradient prototype* $g_{\boldsymbol{W}^{t-2},c}^{pro}$ of class $c$ in the $(t-1)$-th round,[6] which is the averaged gradient of training samples in class $c$ w.r.t. $\boldsymbol{W}^{t-2}$ across selected clients in last round (Shang et al., 2022b):

$$g_{\boldsymbol{W}^{t-2},c}^{pro} = \frac{1}{|\mathcal{C}_c^{t-1}|} \sum_{k \in \mathcal{C}_c^{t-1}} g_{\boldsymbol{W}^{t-2},k,c}^{pro}, \tag{10}$$

$$g_{\boldsymbol{W}^{t-2},k,c}^{pro} = \nabla_{\boldsymbol{W}^{t-2}} L(\boldsymbol{W}^{t-2}; \boldsymbol{P}^{t-2}, \mathcal{D}_{k,c}), \tag{11}$$

where $\mathcal{C}_c^{t-1}$ represents the set of clients sampled in the $(t-1)$-th round and have the training samples of class $c$, and $\mathcal{D}_{k,c}$ denotes all training samples of class $c$ in $\mathcal{D}_k$. Thus, it requires each client sampled in the previous round to first calculate the local gradient prototype of each class $c \in \mathcal{L}_k$ on the same model $(\boldsymbol{P}^{t-2}, \boldsymbol{W}^{t-2})$, return $\{g_{\boldsymbol{W}^{t-2},k,c}^{pro} | c \in \mathcal{L}_k\}$ back to the server along with other local gradients. The server will average and update the global gradient prototypes, and broadcast them in the current round.[7] Based on Eq. (9) and Eq. (10), the gradients for re-balancing $\boldsymbol{W}_k^{i-1}$ are

$$g_{\boldsymbol{W}_k^{i-1}}^{bal} = \frac{1}{|\mathcal{L}|} \left( \sum_{c \in \mathcal{L}_k^{bal}} g_{\boldsymbol{W}_k^{i-1},c}^{bal} + \sum_{c \in \mathcal{L} \setminus \mathcal{L}_k^{bal}} g_{\boldsymbol{W}^{t-2},c}^{pro} \right), \tag{12}$$

where $\mathcal{L}_k^{bal} \subset \mathcal{L}_k$ is the label set in which each class contains at least $T$ samples and $\mathcal{L}$ is the entire label set.

**Updating $\boldsymbol{W}$.** According to the form of Eq. (6), the final updating rule for $\boldsymbol{W}_k^{i-1}$ is[8]

$$\boldsymbol{W}_k^i = \boldsymbol{W}_k^{i-1} - \eta_l \left[ g_{\boldsymbol{W}_k^{i-1}}^{local} + \lambda g_{\boldsymbol{W}_k^{i-1}}^{bal} (\|g_{\boldsymbol{W}_k^{i-1}}^{local}\| / \|g_{\boldsymbol{W}_k^{i-1}}^{bal}\|) \right], \tag{13}$$

In Eq. (13), we normalize the scale of $g_{\boldsymbol{W}_k^{i-1}}^{bal}$ at each step, by making its scale consistent with the decreasing trend of the scale of real gradients during training. The reason is, $g_{\boldsymbol{W}^{t-2},c}^{pro}$ is a **constant** used to calculate $g_{\boldsymbol{W}_k^{i-1}}^{bal}$ and update $\boldsymbol{W}_k^{i-1}$, while the scale of local gradients $g_{\boldsymbol{W}_k^{i-1}}^{local}$ **decreases** during the training, **it will do harm to the training when $g_{\boldsymbol{W}^{t-2},c}^{pro}$ becomes the dominant part on updating the model for consecutive steps**.

After training, the new model is $(\boldsymbol{P}_k^t, \boldsymbol{W}_k^t, \widehat{\boldsymbol{W}}_k^t)$, and client $k$ sends the local gradients $(g_{\boldsymbol{P}^{t-1},k}, g_{\boldsymbol{W}^{t-1},k}, g_{\widehat{\boldsymbol{W}}^{t-1},k}) = (\boldsymbol{P}_k^t - \boldsymbol{P}^{t-1}, \boldsymbol{W}_k^t - \boldsymbol{W}^{t-1}, \widehat{\boldsymbol{W}}_k^t - \widehat{\boldsymbol{W}}^{t-1})$ along with the local gradient prototypes $\{g_{\boldsymbol{W}^{t-1},k,c}^{pro} | c \in \mathcal{L}_k\}$ to the server.

### 3.3.2 SERVER AGGREGATION STAGE

The server first aggregates the gradients and updates the global model as

$$(\boldsymbol{P}^t, \boldsymbol{W}^t, \widehat{\boldsymbol{W}}^t) = (\boldsymbol{P}^{t-1}, \boldsymbol{W}^{t-1}, \widehat{\boldsymbol{W}}^{t-1}) - \eta_s \sum_{k \in \mathcal{C}^t} \frac{|\mathcal{D}_k|}{\sum_{i \in \mathcal{C}^t} |\mathcal{D}_i|} (g_{\boldsymbol{P}^{t-1},k}, g_{\boldsymbol{W}^{t-1},k}, g_{\widehat{\boldsymbol{W}}^{t-1},k}). \tag{14}$$

Also, the server needs to update the global gradient prototypes as

$$g_{\boldsymbol{W}^{t-1},c}^{pro} = \begin{cases} \frac{1}{|\mathcal{C}_c^t|} \sum_{k \in \mathcal{C}_c^t} g_{\boldsymbol{W}^{t-1},k,c}^{pro}, & \mathcal{C}_c^t \neq \emptyset, \\ g_{\boldsymbol{W}^{t-2},c}^{pro}, & \mathcal{C}_c^t = \emptyset, \end{cases} \tag{15}$$

---

[6]We discuss about the impact of utilizing previous gradients information in current round in Appendix A.

[7]We discuss about the limitation of extra communication cost caused by transmitting global gradient prototypes in Appendix B, and a privacy-preserving manner to transmit them in the Ethics Statement Section.

[8]$\min_{\boldsymbol{W}_P} L(\boldsymbol{W}_P; \boldsymbol{P}, \mathcal{D}^{bal})$ is a constant when taking derivatives over $\boldsymbol{W}$.

in which the second case represents that all clients in $\mathcal{C}^t$ in current round do not contain samples of class $c$. In this case, we re-use the global gradient prototype of class $c$ from the previous round. The updated global model and global gradient prototypes are sent to the sampled clients in the next round.

### 3.4 INFERENCE STAGE

After federated training, we only keep the re-balanced classifier $\boldsymbol{W}$ and abandon $\widehat{\boldsymbol{W}}$ during inference:

$$y_{\text{pred}} = \arg \max_i [\boldsymbol{W}^T f(x; \boldsymbol{P})]. \tag{16}$$

## 4 EXPERIMENTS AND ANALYSIS

### 4.1 EXPERIMENTAL SETTINGS

**Datasets and Models**  We conduct experiments on three popular image classification benchmarks: MNIST (LeCun et al., 1998), CIFAR-10 and CIFAR-100 (Krizhevsky et al., 2009). We follow existing studies (Cao et al., 2019; Shang et al., 2022b) to create the long-tailed versions of training sets of above three datasets (i.e., MNIST-LT, CIFAR-10/100-LT), and keep the test sets as balanced. We first define the term *Imbalance Ratio*: IR $= \frac{\max_c \{n_c\}}{\min_c \{n_c\}}$, which is the ratio between the maximum sample number across all classes and the minimum sample number across all classes, to reflect the imbalance degree of the global data distribution. Then, the training sample quantity of each class follows an exponential decay. We choose IR $= 10, 50, 100$ in our main experiments. We also conduct experiments in another *binary class imbalance setting* (Shen et al., 2021), the details and results are in Appendix E. Furthermore, we follow the existing studies (Reddi et al., 2020; Shang et al., 2022b) to adopt the Dirichlet distribution $\text{Dir}(\alpha)$ for the non-i.i.d. data partitioning, in which $\alpha$ controls the non-i.i.d. degree. We set $\alpha = 1.0$ in our main experiments, and put the results on other $\alpha$s in Appendix F. We use the convolutional neural network (CNN) (McMahan et al., 2017) for MNIST-LT, and use ResNet-56 (He et al., 2016) for CIFAR-10/100-LT. More details are in Appendix C.1.

**Baseline Methods**  We mainly compare our method with the existing federated long-tailed learning algorithms, including FedAvg with the CrossEntropy Loss (FedAvg+CE) applied in the local training (McMahan et al., 2017), Fed-Focal Loss (Sarkar et al., 2020), Ratio Loss (Wang et al., 2021), CLIMB (Shen et al., 2021), and the state-of-the-art method CReFF (Shang et al., 2022b). Also, we conduct extra experiments to compare with other FL methods that are only designed for tackling the non-i.i.d. data issue (i.e., FedProx (Li et al., 2018), FedAvgM (Hsu et al., 2019) and FedAdam (Reddi et al., 2020)), the results are in Appendix H.

**Training Details**  We conduct experiments in two popular FL settings based on the ratio of clients participating in each round: (1) **Full client participation** setting: all clients participate in updating the global model in each round, and the total number of clients is 10; (2) **Partial client participation** setting: the total number of clients is 50 but only 10 clients are randomly sampled in each round. We also conduct experiments with a larger number of total clients (i.e., 100), the results are in Appendix I. We adopt SGDM as the optimizer for local training. The local learning rate is 0.01 for MNIST-LT and 0.1 for CIFAR-10/100-LT. The number of local epochs is 5 for all datasets. As for our method, the re-balance factor $\lambda$ is fixed as 0.1 in all experiments, and we explore the effect of different values of $\lambda$ in Appendix J. The quantity threshold $T$ for each class to create the local balanced dataset is set as 8 for MNIST-LT and CIFAR-10-LT, and 2 for CIFAR-100-LT, and we put further discussion in Section 5.1. Each experiment is run on 3 random seeds. Complete training details (e.g., the number of communication rounds in each setting, detailed hyper-parameters of baselines) are in Appendix C.2.

### 4.2 MAIN RESULTS

In the main paper, we report the averaged accuracy over the last 10 rounds (with standard deviation) on the balanced testing set in each experiment following Reddi et al. (2020). **We also display the averaged test accuracy on tail classes in each setting in Appendix D to show that our method can significantly bring improvement to the model's performance on tail classes.** The overall results on the balanced testing sets under full client participation setting are in Table 1, and Table 2

Table 1: Results under the **full client participation** setting. We report the overall test accuracy and standard deviation on the balanced testing set of each dataset.

| Method | MNIST-LT | | | CIFAR-10-LT | | | CIFAR-100-LT | | |
|---|---|---|---|---|---|---|---|---|---|
| | IR = 10 | IR = 50 | IR = 100 | IR = 10 | IR = 50 | IR = 100 | IR = 10 | IR = 50 | IR = 100 |
| FedAvg+CE | 97.99($\pm$ 0.02) | 95.98($\pm$ 0.03) | 92.71($\pm$ 0.50) | 76.21($\pm$ 0.76) | 68.41($\pm$ 0.54) | 59.83($\pm$ 0.52) | 49.08($\pm$ 0.22) | 36.47($\pm$ 0.56) | 33.28($\pm$ 0.28) |
| Fed-Focal Loss | 97.90($\pm$ 0.03) | 96.14($\pm$ 0.08) | 92.97($\pm$ 0.64) | 77.92($\pm$ 0.60) | 61.21($\pm$ 1.91) | 59.86($\pm$ 0.49) | 48.14($\pm$ 0.26) | 35.51($\pm$ 0.70) | 30.05($\pm$ 0.81) |
| Ratio Loss | 97.96($\pm$ 0.04) | 96.20($\pm$ 0.04) | 92.99($\pm$ 0.36) | 78.58($\pm$ 0.42) | 68.01($\pm$ 0.48) | 59.27($\pm$ 0.47) | 48.30($\pm$ 0.16) | 37.62($\pm$ 0.30) | 31.92($\pm$ 0.38) |
| CLIMB | 97.89($\pm$ 0.03) | 95.87($\pm$ 0.06) | 92.71($\pm$ 0.47) | 78.95($\pm$ 0.48) | 66.25($\pm$ 0.57) | 57.67($\pm$ 1.06) | 49.27($\pm$ 0.13) | 36.13($\pm$ 0.24) | 32.18($\pm$ 0.35) |
| CReFF | 97.68($\pm$ 0.03) | 96.49($\pm$ 0.03) | 93.85($\pm$ 0.35) | 83.18($\pm$ 0.27) | 73.46($\pm$ 0.36) | 69.36($\pm$ 0.32) | 46.58($\pm$ 0.42) | 35.82($\pm$ 0.67) | 33.46($\pm$ 0.26) |
| Ours | **98.34**($\pm$ 0.02) | **97.06**($\pm$ 0.03) | **95.73**($\pm$ 0.46) | **83.74**($\pm$ 0.20) | **74.01**($\pm$ 0.31) | **71.04**($\pm$ 0.46) | **51.09**($\pm$ 0.13) | **38.49**($\pm$ 0.34) | **34.63**($\pm$ 0.29) |

Table 2: Results under the **partial client participation** setting. We report the overall test accuracy and standard deviation on the balanced testing set of each dataset.

| Method | MNIST-LT | | | CIFAR-10-LT | | | CIFAR-100-LT | | |
|---|---|---|---|---|---|---|---|---|---|
| | IR = 10 | IR = 50 | IR = 100 | IR = 10 | IR = 50 | IR = 100 | IR = 10 | IR = 50 | IR = 100 |
| FedAvg+CE | 95.51($\pm$ 0.20) | 91.82($\pm$ 0.17) | 89.92($\pm$ 0.54) | 60.38($\pm$ 0.36) | 45.15($\pm$ 0.39) | 40.06($\pm$ 0.96) | 40.81($\pm$ 0.19) | 24.62($\pm$ 0.63) | 22.08($\pm$ 0.83) |
| Fed-Focal Loss | 96.79($\pm$ 0.11) | 92.59($\pm$ 0.14) | 90.45($\pm$ 0.42) | 61.16($\pm$ 0.34) | 46.20($\pm$ 0.34) | 41.10($\pm$ 0.82) | 40.85($\pm$ 0.19) | 24.73($\pm$ 0.46) | 20.17($\pm$ 1.31) |
| Ratio Loss | 95.17($\pm$ 0.21) | 91.10($\pm$ 0.26) | 89.64($\pm$ 0.51) | 63.97($\pm$ 0.29) | 44.22($\pm$ 0.31) | 42.11($\pm$ 0.68) | 40.96($\pm$ 0.36) | 24.12($\pm$ 0.76) | 23.06($\pm$ 0.69) |
| CLIMB | 95.67($\pm$ 0.15) | 92.24($\pm$ 0.22) | 89.75($\pm$ 0.36) | 61.75($\pm$ 0.31) | 46.91($\pm$ 0.34) | 42.02($\pm$ 1.03) | 40.64($\pm$ 0.17) | 23.99($\pm$ 0.85) | 21.44($\pm$ 1.18) |
| CReFF | 96.29($\pm$ 0.08) | 94.16($\pm$ 0.19) | 92.16($\pm$ 0.17) | 69.38($\pm$ 0.24) | 60.52($\pm$ 0.21) | 55.63($\pm$ 0.60) | 39.38($\pm$ 0.13) | 25.42($\pm$ 0.24) | 24.77($\pm$ 0.50) |
| Ours | **97.54**($\pm$ 0.06) | **95.17**($\pm$ 0.11) | **93.61**($\pm$ 0.15) | **71.68**($\pm$ 0.35) | **61.42**($\pm$ 0.30) | **57.11**($\pm$ 0.52) | **42.97**($\pm$ 0.11) | **27.73**($\pm$ 0.29) | **25.64**($\pm$ 0.43) |

displays the results under partial client participation setting. We can draw the main conclusion from these tables as: **our method consistently outperforms the existing algorithms in all settings.**

As we can see, Fed-Focal Loss underperforms FedAvg with CE loss in some settings, which validates the claim that directly applying the centralized long-tailed learning methods can not help to address the global class imbalance problem in FL, as it ignores the mismatch between the global and the local imbalance patterns. Ratio Loss and CLIMB apply the class-level re-weighting and client-level re-weighting ideas separately, and gain slight improvement over FedAvg. We analyze that the reason for the limited improvement lies in that though the re-weighting practice helps the model to focus more on the learning of tail classes, it is not conducive to the representation learning on the abundant data of head classes according to Kang et al. (2019). Moreover, the assumption of obtaining a global auxiliary dataset makes Ratio Loss impractical in real cases.

The great performance of CReFF helps to validates the effectiveness of the idea of classifier re-balancing. However, the optimization of the federated features requires massive computations on the server (especially when the number of classes is large), and the federated features from the same class may converge to be similar. Thus, the re-trained classifier faces the problem that it may overfit on the highly similar and small amount of the federated features, which is reflected in the poorer performance under smaller IR. **Our method instead takes full advantage of the local real data supplemented by the global gradient prototypes to locally re-balance the classifier, and consistently outperforms all baselines by a large margin.** Compared with CReFF and Ratio Loss, we do not have extra requirements except for the normal aggregations on the server, and produce a re-balanced classifier that has better generalization ability with the help of abundant real data.

We further display the evaluation accuracy curve after each round in CIFAR-10-LT (IR = 100) under the full client participation setting in Figure 2. As we can see, **our method not only has the best converged performance, but also achieves much faster convergence speed than all baseline methods.** That is because our method re-balances the classifier at each local training step, and this makes it converge faster to the optimal balanced classifier.

## 5 FURTHER EXPLORATIONS

### 5.1 LOCAL REAL DATA PLAYS AN IMPORTANT ROLE IN RE-BALANCING THE CLASSIFIER

During creating the local balanced datasets, we set a threshold $T$ to decide whether each client owns the enough data of a specific class. Larger $T$ decreases the number of classes in which the real data can be used to calculate local gradient prototypes in Eq. (9), while smaller $T$ leads to the relatively unreliable gradients of class $c$. We then explore the effect of different $T$s on CIFAR-10-LT, and put

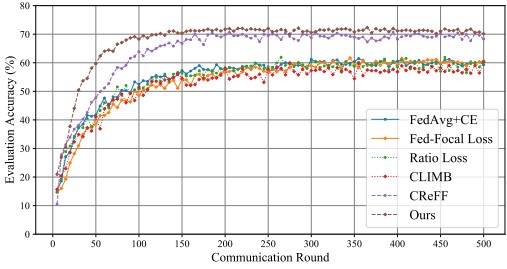 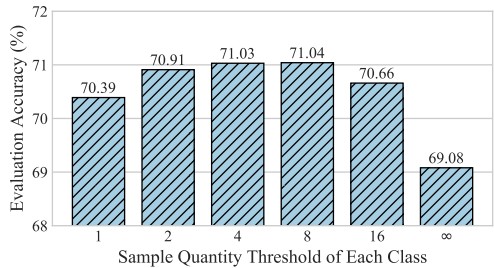

Figure 2: The evaluation accuracy curve of each method in CIFAR-10-LT. Our method achieves faster convergence speed and better performance than all existing baselines.

Figure 3: The test accuracy of using different sample quantity thresholds for each class when creating the local balanced datasets in CIFAR-10-LT with full client participation.

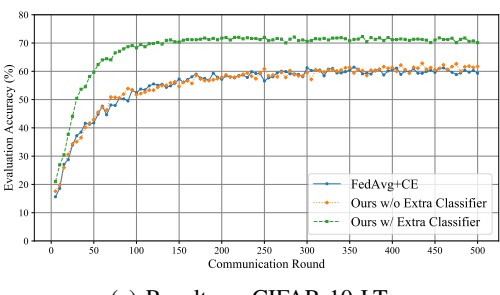 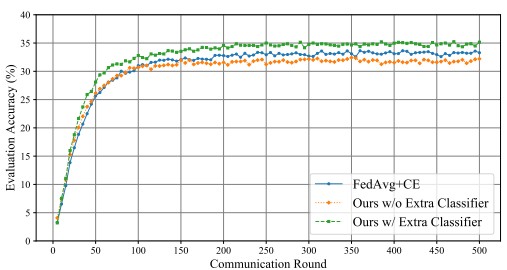

(a) Results on CIFAR-10-LT.

(b) Results on CIFAR-100-LT.

Figure 4: The positive effect of adding a supplementary classifier in the training phase to help model the global data distribution when locally re-balancing the classifier by our method.

the results in Figure 3. We indeed observe a trade-off pattern as expected and find that $T = 4, 8$ are generally proper choices. $T = \infty$ means we remove the role of local real data on the classifier re-balancing and only use the global gradient prototypes instead, and we find the performance degrades greatly, which **verifies the large benefits of using local real data to adjust the classifier**.

### 5.2 SUPPLEMENTARY CLASSIFIER IS CRUCIAL FOR EFFECTIVE CLASSIFIER RE-BALANCING

Here, we conduct experiments on CIFAR-10/100-LT to explore the necessity of introducing a supplementary classifier to address the optimization difficulty caused by performing classifier re-balancing locally. The results are displayed in Figure 4. "FedAvg+CE" is the naive baseline, which can also be considered as an ablation situation in which we add the supplementary classifier $\hat{W}$ but without further re-balancing on $W$. "Ours w/o Extra Classifier" represents the situation where we do not add $\hat{W}$ but still re-balance $W$ based on Eq. (13). We observe that if we do not add the supplementary classifier in the training phase, the model will converge to a bad local optimum and behave much worse than that if we adopt the two-stream classifier architecture. **This helps to validate our motivation and the great effectiveness of introducing a new global classifier to address the optimization difficulty brought by the local classifier re-balancing practice.**

## 6 CONCLUSION

In this paper, we propose a decentralized decoupling mechanism to effectively re-balance the classifier for tackling federated long-tailed learning. Motivated by the distributed characteristic of FL, we propose to re-balance the classifier during local training with the help of abundant local real data supplemented by global gradient prototypes. Furthermore, in order to address the problem of contradictory optimization goals brought by performing local classifier re-balancing, we introduce a two-stream classifiers architecture to help model the global data distribution. Thorough experiments verify the great effectiveness of our method over strong baselines without extra data requirements.

ETHICS STATEMENT

Our purpose is to address the optimization problem of FL on the non-i.i.d. and long-tailed data and help to learn a better global model that has good performance on all classes. The datasets used in our experiments are all publicly available. Also, our method only requires the normal gradients transmission between the server and the clients as other FL methods do, which will not expose the local data privacy. Specifically, when uploading global gradient prototypes, we can use privacy-preserving methods such as Homomorphic Encryption to allow **the server only get the encrypted average of global gradient prototype of each class instead of the single local gradient prototype from each client**, which further enhances the protection of the local privacy. The steps are the following:

- Create a secret key that is only known by clients.
- Clients encrypt the gradients and the gradient prototypes with the secrete key, then upload them to the server.
- The server only calculate the homomorphic average of the local gradients and gradient prototypes, but can not obtain the original values of gradient information as it does not know the secrete key. Then, the server broadcasts the updated information to the clients in the next rounds.
- After receiving the encrypted global information from the server, the clients decrypt the information in the local with the key, and use it to perform the local training.

REPRODUCIBILITY STATEMENT

We include all the details for reproducing the results in this work in both the main paper and the appendix. We present the code source and the infrastructure used in our experiments at the end of Appendix C.2. We introduce all necessary experimental settings (such as datasets, models, baseline methods, hyper-parameters for each method, all training details) in Section 4.1 and Appendix C. All datasets used in our experiments all publicly available. We will release the code upon acceptance.

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

## A    THE IMPACT OF UTILIZING PREVIOUS GRADIENT INFORMATION ON RE-BALANCING CURRENT CLASSIFIER

In our proposed method, it requires to use the global gradient prototypes calculated on the previous classifier $W^{t-2}$ to help re-balancing current classifier $W^{t-1}$. This may cause some inconsistency with the ideal solution that calculates the global gradient prototypes w.r.t. $W^{t-1}$. However, due to the constraint in FL that the client can only get the information from last round at the beginning of current round, it is unavoidable to use previous gradient prototypes in current round, thus we make this adjustment to make classifier re-balancing feasible. Furthermore, **using gradient information from previous rounds to help current round's model training is widely adopted in previous studies (Karimireddy et al., 2020b; Shang et al., 2022b) with both theoretical and great empirical guarantees. Our experimental results also validate the empirical effectiveness of this approach.**

## B    THE EXTRA COMMUNICATION COST CAUSED BY TRANSMITTING GRADIENT PROTOTYPES BETWEEN THE SERVER AND CLIENTS

In our method, besides uploading the normal gradient of the local model, each selected client also needs to upload the additional local gradient prototypes w.r.t. the classifier $W$ and the gradient w.r.t. the additional classifier $\hat{W}$ to the server. Also, the server will then broadcast the aggregated global gradient prototypes to the clients in the following rounds. This will cause extra communication costs between the server and clients compared with some baselines such as Ratio Loss and CLIMB, which we admit is the potential limitation of our method. However, **we think the extra communication cost and storage memory of the supplementary classifier and global gradient prototypes are the necessary costs to effectively improve global model's performance on long-tailed global data.** Previous FL studies (Karimireddy et al., 2020b; Shang et al., 2022b; Karimireddy et al., 2020a) have proven that introducing extra gradient information in the local training can bring significant improvement, thus the achievement gained by these advanced methods is also caused by adding extra gradients and parameters. For instance, SCAFFOLD (Karimireddy et al., 2020b) requires an additional transmission of the local control variate for each client that has the same number of parameters of the original model. Furthermore, **compared with previous state-of-the-art method CReFF (Shang et al., 2022b), our method does not introduce extra parameters/gradients, while consistently achieves better performance.** Finally, the total number of these newly introduced parameters is acceptable compared with the original model/gradient parameters. Take CIFAR-100 with 100 classes for example, the number of newly introduced gradient parameters is about $2 \times 10^6$, which is comparable with the original based model CIFAR-56 with $6 \times 10^5$ parameters, and is much smaller than CIFAR-50 with $2 \times 10^7$ parameters. All in all, the extra communication costs are worthwhile for improving model's performance.

## C    DETAILED EXPERIMENTAL SETTINGS

### C.1    DATASETS AND MODELS

Here, we introduce the datasets and the backbone models we used in our experiments. We choose three classical image classification tasks, including MNIST[9] (LeCun et al., 1998), CIFAR-10 and CIFAR-100[10] (Krizhevsky et al., 2009). We then follow existing centralized and federated long-tailed learning studies (Cao et al., 2019; Shang et al., 2022b) to create the long-tailed versions of the training sets of above datasets (i.e., MNIST-LT, CIFAR-10/100-LT). Specifically, the long-tailed degree is controlled by a ratio called the *Imbalance Ratio*: IR$=\frac{\max_c\{n_c\}}{\min_c\{n_c\}}$, where $n_c$ represents the sample quantity of class $c$ (0-indexed). Then, we manage to make the sample quantity of each class follow an exponential decay trend:

$$n_c = n_0 \times \left(\frac{1}{\text{IR}}\right)^{\frac{c}{C-1}}, \quad c = 0, \cdots, C-1. \tag{17}$$

---

[9]Can be downloaded from http://yann.lecun.com/exdb/mnist/.
[10]Can be downloaded from https://www.cs.toronto.edu/ kriz/cifar.html.

As for the non-i.i.d. data partitioning, we follow existing studies (Reddi et al., 2020; Shang et al., 2022b) to adopt the Dirichlet distribution $\text{Dir}(\alpha)$. Smaller $\alpha$ means the heavier non-i.i.d. degree. We choose $\alpha = 1.0$ in our main paper, and we also put the results on different $\alpha$s in Appendix F.

We use the same convolutional neural network (CNN) used in McMahan et al. (2017) for experiments on MNIST-LT, and adopt ResNet-56 (He et al., 2016) as the backbone model for CIFAR-10/100-LT.

## C.2 COMPLETE TRAINING DETAILS

### LOCAL TRAINING SETTINGS

We utilize SGDM as the local optimizer in all experiments. The search grid for server learning rate is $[0.5, 1.0, 2.0, 3.0]$, and the search grid for local learning rate is $[0.01, 0.05, 0.1, 0.5]$. The final local learning rate is 0.01 for MNIST-LT, and 0.1 for CIFAR-10/100-LT. The tuned server learning rate is 1.0 for all settings. For all three datasets, the batch size for local training is 64, and the number of local training epochs is 5. As mentioned in main paper, we perform experiments in both full client participation and partial client participation settings. We set different total communication rounds in different setting considering the different convergence speeds of the global models: (1) In the full client participation setting, the number of communication rounds is 200 for MNIST-LT, and 500 for CIFAR-10/100-LT. (2) In the partial client participation setting, the number of communication rounds is 500 and 1000 for MNIST-LT and CIFAR-10/100-LT separately.

### SERVER AGGREGATION SETTINGS

During server aggregation, we follow the same procedure as that in FedAvg to aggregate the collected local gradients in the current round, and update the global model with the averaged gradients. The server learning rate is tuned as 1.0 for all experiments. Furthermore, as for CReFF and our method, the server needs to update the global gradient prototypes (refer to Section 3.3 in our main paper) by averaging local gradient prototypes. However, different from CReFF, we do not have extra requirements on the server to make it perform further optimization and training process.

### HYPER-PARAMETERS OF EACH METHOD

Here, we introduce the choices of hyper-parameters used in each method in detail.

**Fed-Focal Loss:** Fed-Focal Loss (Sarkar et al., 2020) directly applies Focal Loss (Lin et al., 2017) to the local training. The form of Focal Loss is

$$L_{focal} = -(1 - p_T)^\gamma \log(p_T), \tag{18}$$

where $p_T$ is the predicted probability of the sample corresponding to the ground truth class. We set $\gamma = 2$ in our experiments.

**Ratio Loss:** Ratio Loss (Wang et al., 2021) applies the class-level re-weighting practice by first estimating the global imbalance pattern on the server with an auxiliary balanced dataset. Its form can be written as

$$L_{ratio} = (\alpha + \beta\mathbb{R})L_{CE}, \tag{19}$$

where $L_{CE}$ is the traditional CrossEntropy Loss, $\mathbb{R}$ is the ratio vector that contains the relatively estimated sample quantity of each class on the server, $\alpha$ and $\beta$ are two hyper-parameters. Thus, we follow the original study (Wang et al., 2021) to set the sample number of each class on the auxiliary balanced dataset to be 32, $\alpha = 1.0$, $\beta = 0.1$.

**CLIMB:** CLIMB (Shen et al., 2021) aims to perform the client-level re-weighting to up-weight the aggregation weights for the local gradients with larger local training losses, as the global model behaves poorly on these clients' local data. The hyper-parameters in CLIMB include a tolerance constant $\epsilon$ and a dual step size $\eta_D$. In our experiments, we follow the original setting to set $\epsilon = 0.01$ for MNIST-LT and $\epsilon = 0.1$ for CIFAR-10/100-LT, set $\eta$ as 2.0 and 0.1 for MNIST-LT and CIFAR-10/100-LT separately.

Table 3: Averaged evaluation accuracy on the tail classes under the **full client participation** setting in main experiments.

| Method | MNIST-LT | | | CIFAR-10-LT | | | CIFAR-100-LT | | |
|---|---|---|---|---|---|---|---|---|---|
| | IR $= 10$ | IR $= 50$ | IR $= 100$ | IR $= 10$ | IR $= 50$ | IR $= 100$ | IR $= 10$ | IR $= 50$ | IR $= 100$ |
| FedAvg+CE | 96.05 | 90.23 | 82.21 | 67.87 | 55.26 | 29.24 | 36.25 | 13.37 | 8.50 |
| Fed-Focal Loss | 95.84 | 90.47 | 82.62 | 74.94 | 47.37 | 33.86 | 34.40 | 12.53 | 6.49 |
| Ratio Loss | 96.04 | 90.76 | 83.01 | 71.27 | 55.79 | 33.28 | 34.55 | 15.35 | 7.98 |
| CLIMB | 95.70 | 89.93 | 82.01 | 73.68 | 55.86 | 30.95 | 35.81 | 13.31 | 8.65 |
| CReFF | 95.98 | 91.98 | 86.62 | 82.87 | 66.10 | 57.03 | 38.18 | **22.77** | **18.98** |
| Ours | **96.78** | **92.97** | **89.59** | **83.86** | **69.74** | **60.41** | **40.11** | 20.19 | 15.58 |

Table 4: Averaged evaluation accuracy on the tail classes under the **partial client participation** setting in main experiments.

| Method | MNIST-LT | | | CIFAR-10-LT | | | CIFAR-100-LT | | |
|---|---|---|---|---|---|---|---|---|---|
| | IR $= 10$ | IR $= 50$ | IR $= 100$ | IR $= 10$ | IR $= 50$ | IR $= 100$ | IR $= 10$ | IR $= 50$ | IR $= 100$ |
| FedAvg+CE | 89.75 | 80.05 | 74.35 | 51.99 | 24.03 | 2.64 | 29.65 | 8.55 | 5.74 |
| Fed-Focal Loss | 92.65 | 82.43 | 75.46 | 49.78 | 27.43 | 5.68 | 29.51 | 8.89 | 4.08 |
| Ratio Loss | 89.83 | 77.93 | 73.35 | 55.54 | 23.81 | 4.87 | 29.28 | 8.36 | 5.86 |
| CLIMB | 90.09 | 81.49 | 74.58 | 54.45 | 23.90 | 4.14 | 29.85 | 8.37 | 4.97 |
| CReFF | 92.87 | 88.94 | 83.75 | 74.81 | 62.67 | **45.30** | 34.38 | 17.46 | **16.58** |
| Ours | **94.82** | **90.06** | **86.38** | **75.71** | **63.58** | 43.92 | **35.34** | **17.91** | 13.39 |

**CReFF:** CReFF (Shang et al., 2022b) needs to create a set of federated features on the server, of which the number per class is 100. Following the original setting, the optimization steps on the federated features is 100, the classifier re-training steps is 300. Further, the learning rate of optimizing the federated features is 0.1 for all datasets, and the learning rate of classifier re-training is kept as the same as that used in the local training on that dataset.

**RedGrape (Ours):** Our method introduces two hyper-parameters: $\lambda$ for the classifier re-balancing strength, and $T$ for the sample quantity threshold of each class on creating local balanced datasets. We put the detailed discussions about these two hyper-parameters in our main paper. The recommended search grids for $\lambda$ are $\{1.0, 0.1, 0.01\}$, and $\{2, 4, 8\}$ for $T$.

CODE AND INFRASTRUCTURE

Our code is implemented based on the open-sourced FL platform FedML (He et al., 2020). We will release our code upon acceptance. Our experiments are conducted on 8 * GeForce RTX 2080 Ti.

## D   RESULTS ON THE TAIL CLASSES IN MAIN EXPERIMENTS

In our main paper, we put the results of the overall accuracy on the balanced testing sets of each method, and we have that **our method consistently outperform all other methods in all settings.** Here, we put the averaged accuracy on the tail classes of each method. Specifically, we define the tail classes as the last 30% classes with the minimum sample quantity.

The results on the tail classes are in Table 3 and Table 4. As we can see, **our method brings significant improvement on the tail classes in most cases.** Also, we find that CReFF tends to achieve better performance on the tail classes when the imbalance degree is larger. However, the performance of CReFF on the overall testing sets is worse than our method according to the results in our main paper. This validates our analysis that, re-training the classifier on a set number of federated features on the server can indeed re-balance the classifier to some extent, but it is likely to produce a sub-optimal classifier that overfits on these limited number of pseudo features.

Table 5: The overall evaluation accuracy in the binary class imbalance setting. IR = 100 and the randomly chosen tail classes are 0, 7 and 8.

| Method | Full Participation | | Partial Participation | |
|---|---|---|---|---|
| | MNIST | CIFAR-10 | MNIST | CIFAR-10 |
| FedAvg+CE | 93.91 | 70.91 | 91.97 | 61.27 |
| Fed-Focal Loss | 93.53 | 70.77 | 91.77 | 59.18 |
| Ratio Loss | 94.23 | 73.22 | 92.32 | 62.86 |
| CLIMB | 93.89 | 72.04 | 92.11 | 63.18 |
| CReFF | 96.70 | 78.98 | 96.01 | 71.41 |
| Ours | **96.86** | **79.88** | **96.43** | **73.31** |

Table 6: The overall accuracy on the balanced testing sets under different non-i.i.d. degrees. Our method consistently outperforms all baselines.

| Method | MNIST-LT | | | CIFAR-10-LT | | | CIFAR-100-LT | | |
|---|---|---|---|---|---|---|---|---|---|
| | $\alpha = 0.1$ | $\alpha = 1.0$ | $\alpha = 10.0$ | $\alpha = 0.1$ | $\alpha = 1.0$ | $\alpha = 10.0$ | $\alpha = 0.1$ | $\alpha = 1.0$ | $\alpha = 10.0$ |
| FedAvg+CE | 93.42 | 92.71 | 94.09 | 59.46 | 59.83 | 63.01 | 31.18 | 33.28 | 31.74 |
| Fed-Focal Loss | 94.16 | 92.97 | 94.21 | 53.47 | 59.86 | 61.06 | 31.72 | 30.05 | 30.18 |
| Ratio Loss | 93.50 | 92.99 | 94.34 | 59.30 | 59.27 | 61.82 | 32.60 | 31.92 | 31.93 |
| CLIMB | 93.80 | 92.71 | 94.23 | 61.10 | 57.67 | 61.46 | 31.64 | 32.18 | 32.45 |
| CReFF | 93.94 | 93.85 | 94.76 | 63.25 | 69.36 | 70.30 | 32.20 | 33.46 | 31.60 |
| Ours | **95.34** | **95.73** | **95.93** | **64.30** | **71.04** | **71.82** | **32.86** | **34.63** | **34.42** |

## E    RESULTS IN ANOTHER CLASS IMBALANCE SETTING

We also conduct experiments in a *binary class imbalance* setting in FL (Shen et al., 2021), in which three classes are randomly chosen as the tail classes, and they are assigned with 1/IR number of sampled compared with other normal/head classes. The experiments are conducted on MNIST-LT and CIFAR-10-LT datasets with IR = 100, and other experimental settings are kept as the same as that in our main experiments. The results of the overall evaluation accuracy are in Table 5. The conclusion remains the same that, **our method achieves the best performance in all cases.**

## F    EXPERIMENTS UNDER DIFFERENT NON-I.I.D. DEGREES

In our main experiments, we fix the non-i.i.d. degree $\alpha = 1.0$, in order to mainly explore the effects of different imbalance degrees. Here, we conduct extra experiments with $\alpha = 0.1$ and $\alpha = 10.0$, and fix the imbalance ratio IR = 100. We conduct experiment on MNIST-LT and CIFAR-10 under the full client participation setting, and other experimental settings are kept as the same as that in our main experiments. We put the results of the overall evaluation accuracy on the balanced testing sets in Table 6. We can draw the main conclusion from the table that, **our method can achieve the best performance under different non-i.i.d. degrees**.

## G    THE COMPARISON BETWEEN THE JOINT OPTIMIZATION PARADIGM AND THE TWO-STAGE OPTIMIZATION PARADIGM

In our proposed method, we choose to re-balance the classifier during the local training. That is, the representation learning and the classifier re-balancing are performed jointly, which is different from the decoupled training method (Kang et al., 2019) in the centralized setting. The reason is, unlike the centralized setting, the data is only kept in the local clients in FL. Thus, in practical, there is no global balanced dataset on the server to perform the two-stage training by re-training the classifier after representation learning. CReFF (Shang et al., 2022b) then creates some pseudo features in the

Table 7: The comparison between the joint optimization paradigm adopted in our main paper and an alternative two-stage optimization paradigm. IR = 100.

| Method | Full Participation | | Partial Participation | |
|---|---|---|---|---|
| | CIFAR-10-LT | CIFAR-100-LT | CIFAR-10-LT | CIFAR-100-LT |
| Two-Stage Optimization | 65.74 | 32.38 | 52.28 | 21.85 |
| Joint Optimization (in paper) | **71.04** | **34.63** | **57.11** | **25.64** |

Table 8: The overall evaluation accuracy of more FL baselines on MNIST-LT and CIFAR-10-LT with IR = 100 under full client participation setting.

| Method | MNIST-LT | CIFAR-10-LT |
|---|---|---|
| FedAvg+CE | 92.71 | 59.83 |
| FedProx | 92.43 | 60.79 |
| FedAvgM | 94.51 | 57.81 |
| FedAdam | 95.06 | 58.45 |
| Fed-Focal Loss | 92.97 | 59.86 |
| Ratio Loss | 92.99 | 59.27 |
| CLIMB | 92.71 | 57.67 |
| CReFF | 93.85 | 69.36 |
| Ours | **95.73** | **71.04** |

server for re-training the classifier after each round's aggregation. However, we point out that the low quality and limited number of pseudo features only produce a sub-optimal classifier. Therefore, **we are encouraged to put the classifier re-balancing along with the local training by fully utilizing the local real data, and propose such a joint optimization paradigm**.

Also, there exists another solution to locally re-balance the classifier: **train the entire model using standard FedAvg for sufficient rounds, then freeze the encoder and only re-train the classifier under FedAvg framework.** Therefore, we conduct extra experiments with this choice by freezing the encoder trained by the baseline FedAvg and further re-training a new balanced classifier following our RedGrape mechanism with extra 200 rounds until converged. The results in Table 7 suggest that **this two-stage training choice with more communication costs still underperforms the joint optimization paradigm adopted in our main paper**, validating the effectiveness of our original motivation. We think the reason is, Eq. (6) formulates an integrated optimization target that requires synchronous updates for all parameters, while this separate two-stage training practice will lead to a sub-optimal solution. Moreover, above practice will take more communication rounds for extra classifier re-training, which will cause an unfair comparison with other baselines.

## H    RESULTS ON FL METHODS ONLY DESIGNED FOR NON-I.I.D. DATA ISSUE

In our main experiments, we choose the methods that are specifically designed for tackling long-tailed global data distribution as baselines. Here, we make a further comparison between our method with popular FL methods that are designed only for the non-i.i.d. data issue, including FedProx (Li et al., 2018), FedAvgM (Hsu et al., 2019) and FedAdam (FedOpt) (Reddi et al., 2020). We perform experiments on MNIST-LT and CIFAR-10-LT with IR = 100 under the full client participation setting. The results are in Table 8. As we can see, **these heterogeneous data-oriented methods consistently underperform our method, due to the reason that they do not take the long-tailed data issue into consideration and will also produce bad global models that are biased to the head classes.**

Table 9: The overall evaluation accuracy on MNIST-LT and CIFAR-10-LT with $IR = 100$ under full client participation setting, where the total number of clients is 100 and the number of clients sampled in each round is 10.

| Method | MNIST-LT | CIFAR-10-LT |
|---|---|---|
| FedAvg+CE | 87.06 | 32.68 |
| Fed-Focal Loss | 87.56 | 33.04 |
| Ratio Loss | 87.70 | 33.60 |
| CLIMB | 87.46 | 34.38 |
| CReFF | 90.56 | 42.25 |
| Ours | **92.03** | **43.78** |

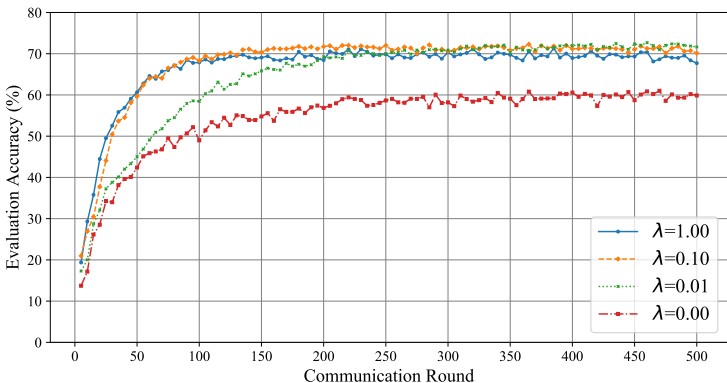

Figure 5: The test accuracy of using different re-balance factor $\lambda$ in CIFAR-10-LT. Smaller $\lambda$ leads to slower convergence speed but relatively better generalization ability.

## I    EXPERIMENTAL RESULTS WITH 100 CLIENTS

In the partial client participation setting in the main paper, the total number of clients is set as 50. In this section, we perform experiments with larger number of total clients. Specifically, we set the total number of clients as 100 and the number of clients sampled in each round is 10. The total number of communication rounds is 500 for MNIST-LT and 1000 for CIFAR-10-LT. The results displayed in Table 9 show that **our method can still achieve significant improvement when the number of clients is larger.**

## J    RE-BALANCING STRENGTH DECIDES ON THE CONVERGENCE TRADE-OFF

In order to solve the optimization target of Eq. (5), we consider to turn Eq. (5) into an unconstrained optimization problem by adding a penalty term on the local objective function as Eq. (6) and update $W$ as Eq. (13), where a re-balance factor $\lambda$ is used to control the re-balancing strength. Here, we conduct experiments to explore the effect of different $\lambda$s on the model's performance, and the results are shown in Figure 5. We find that **the smaller $\lambda$ results in slower convergence speed but obtains relatively better performance of the converged model**. We analyze the reason lies in that, the $g^{bal}_{W^{i-1}_k}$ contains a part of global gradient prototypes calculated in the previous round and is a constant when updating $W$. It will adversely affect the model's convergence in the late stage of the training when we are still using a large $\lambda$ to re-balance the classifier. An interesting direction to improve our method is designing an adaptive $\lambda$ that decays along with the training, which we leave to future work. When $\lambda = 0.0$, the addition of two classifiers equals to one normal classifier used in FedAvg, so FedAvg is a special case of our method in this case.

