# OpenReview forum: "Decentralized Decoupled Training for Federated Long-Tailed Learning"
_ICLR.cc/2024/Conference — Submitted to ICLR 2024_

### Official Review · Reviewer_2RDR · 2023-10-28

**Soundness:** 3 good
**Presentation:** 2 fair
**Contribution:** 2 fair
**Rating:** 5
**Confidence:** 4

**Summary:**

This paper proposes a training algorithm to address the challenge of long-tailed data distribution in federated learning. They target the contradictory optimization goal when training the model and introduce an additional classifier to make it more feasible. Moreover, they decouple the training of the original and additional classifier to make the original classifier benefit from the balanced data gradient. Empirical experiments are conducted to verify the effectiveness of their method.

**Strengths:**

- Originality: The paper focuses on a conventional problem, but the method that introduces an additional classifier to address the contradictory optimization is novel.
  - Quality: Sufficient technique details.
  - Clarity: The logic and presentation are relatively good.
  - Significance: Maybe there is something that I misunderstand in their method. But from my point of view, the method is counterintuitive, and the experiment results are not significant. No theoretical guarantee is provided. These make the significance of this paper limited.

**Weaknesses:**

- The proposed method is somehow counterintuitive. Please check the questions below.
  - The improvement is not so impressive. E.g., in Tab.1 and 2, the maximal improvement is about 1% and 2%, respectively. For some small datasets like MNIST and CIFAR10, the improvement is especially limited. I encourage the authors to discuss more advantages of their paper than the SOTA. Though the author claims CReFF needs more computation, it seems to me that the computation and communication overhead of the proposed method is also high. I expect the author to analyze the difference in the computation cost between theirs and other methods.
  - Given the experiment results are not so good, no theoretical guarantee makes the significance and contribution of this paper limited

**Questions:**

1. In Eq.6, how to calculate the gradient of $\min_{W_P}L(W_P;P,D^{bal})$? From my understanding, in Alg.1, line 3 is to optimize $L(P,W,\hat{W};D_k)$ while line 4-6 can be considered as jointly optimize $L(P,W,\hat{W};D_k)+\lambda L(W;P,D^{bal})$. But I don’t see how to optimize $-\min_{W_P}L(W_P;P,D^{bal})$ where $P$ should be the optimization variable.
  2. Even though I know the whole training process as shown in Alg.1 and Fig.1, I am still confused about the effect of $\hat{W}$. Except that $\hat{W}$ will change the optimization goal as in Eq.5, I cannot get the point that what $\hat{W}$ can really learn in the training process. My intuition is that $\hat{W}$ will learn a residual of the output of a classifier trained with the long-tailed distribution and balanced distribution. But it makes no sense to me to introduce $\hat{W}$ here. We can solely train $W$ since the inference only involves $W$ but $\hat{W}$. Moreover, I notice that the author shows the results in Fig.4 to show the effectiveness of the additional classifier. Can you provide more intuition or explanation about why the additional classifier leads to better performance from a learning perspective?
  3. Why $g_{W^{t-2},c}^{pro}$ is a constant? I think it depends on $W^{t-2}$ and $P^{t-2}$. So I don’t quite get the point why we need to normalize the scale of $g_{W_k^{i-1}}^{bal}$ in Eq.13.
  4. In Eq.9-11, it requires at least one client to have sufficient samples for each class. I doubt the applicability of such a requirement in a realistic scenario. What if the data distribution is a heavy long-tailed distribution, e.g., only a few samples for some classes? In this case, it is very possible that no clients have sufficient samples, which makes their method fail or makes it very hard to tune the threshold $T$. Even though in Fig.3 it seems that the proper choice of $T$ is 2-8, I still think it is too small for sufficient samples and encourage the author to increase $T$ to see whether a larger $T$ can lead to a better performance. Also, when $T=\infty$, no clients have sufficient samples and all the local real data are removed, then how to get the global gradient prototype?
  5. I don't quite understand why Eq.3 is a contradictory target. Because from the formulation of Eq.3, $P, W$ can be achieved to satisfy both $\min L(P,W;\cup_k D_k)$, and $\min L(W;P,D^{bal})$. It is not contradictory based on the math formulation.

---

### Official Review · Reviewer_TDJi · 2023-10-31

**Soundness:** 2 fair
**Presentation:** 1 poor
**Contribution:** 3 good
**Rating:** 3
**Confidence:** 4

**Summary:**

This paper addresses a key challenge in FL when data samples have a long-tailed distribution. In such cases, FL tends to produce models biased towards the majority classes. This paper looks into "decoupled training", a method previously effective in centralized long-tailed learning.

This paper proposes a decentralized decoupled training method. Instead of relying on pseudo features, this method utilizes real data from local storage and combines it with global gradient prototypes. This process creates local balanced datasets to recalibrate the classifier locally. To further enhance the model's performance, this paper also introduces a supplementary classifier. This classifier aids in modeling the overall data distribution, mitigating potential local optimization conflicts.

Empirical experiments demonstrate that this approach consistently outperforms existing methods in various settings.

**Strengths:**

1. The problem studied in this paper is interesting and important. The method addresses the problem of long-tailed which is important in FL.

2. This paper has originality and some technically sound.

**Weaknesses:**

1. The clarity and organization of the paper can be improved. There is a need for clearer motivation for the proposed method, providing a stronger rationale for its development.

2. The performance of the decentralized decoupled training mechanism heavily relies on the abundant local real data, limiting its applications to more wide areas.

**Questions:**

1. There are certain issues with the formulas and symbols used throughout the paper, causing confusion. Specifically,

   - In Eqs (4), (5), and (6), what distinguishes $L(P,W,\widehat{W};\mathcal{D})$ from $L(W;P,\mathcal{D})$? It would be clearer to use distinct symbols for differentiation.

   - In Eqs (7), (8), (9), and (11), the subscripts for the differentiation symbols appear to be incorrect. The correct notation should be $x^{t}=x^{t-1}-\eta\nabla_xf(x^{t-1})$.

   - Eqs (4) and (5) raise some queries: How does $L(W;P,\mathcal{D}^{bal})$ constrain $L(P,W,\widehat{W};\mathcal{D})$? Does the expression $L(W;P,\mathcal{D}^{bal})=\min_{W_P}L(W_P;P,\mathcal{D}^{bal})$ imply that $W=\arg\min_{W_P}L(W_P;P,\mathcal{D}^{bal})$?

2. This paper seems to rely on an implicit assumption that the real data, which is exclusively stored on the clients, is abundant. Such an assumption seems strong, especially in cross-device contexts where locally stored data is typically limited.

3. The Algorithm focuses solely on the local training process. Incorporating a global process for clarity, which includes both local training and global aggregation, would be beneficial.
4. The description, specifically for each class $c$, in point (1) is some ambiguous. Are points (1) and (2) referring to different cases?
5. For the final logits given as $z=W^\top h+ \widehat{W}^\top h$, is there a need for an additional normalization coefficient to satisfy the properties of logits?
6. It would be beneficial if the authors could provide a more comprehensive discussion in the experimental section regarding the additional communication overhead of the proposed method. This encompasses the transmission of extra model parameters or gradient information.

---

### Official Review · Reviewer_EUxj · 2023-10-31

**Soundness:** 3 good
**Presentation:** 3 good
**Contribution:** 2 fair
**Rating:** 5
**Confidence:** 4

**Summary:**

This work studies Federated Learning in a setting where the global data distribution is long-tailed and non-IID decentralized regarding the classes. It adopts the popular decoupled training strategy used in centralized long-tailed training. Each client re-balance the
classifier on a locally balanced dataset based on the local real data and the global gradient prototypes of a supplementary classifier that captures the global distribution.

**Strengths:**

* The writer-up is clear and organized
* The proposed approach is sound and intuitive

**Weaknesses:**

* The scope of this paper is a bit restricted in terms of either imbalanced learning or federated learning (FL). On the imbalanced learning side, the decoupling classifier strategy is a subset of the SOTA techniques in imbalanced learning. On the FL side, imbalanced learning is a subproblem in general non-IID FL.

* The technical contributions are a bit small. It seems to be an application of CReFF and prototyping, which are actually not specific to FL but are already widely used in centralized learning. The proposed method seems to be incremental on CReFF.

* The related work and compared baselines are quite outdated for both centralized imbalanced learning or FL. Previous works in section 2.2 are before 2021. For FL baselines in experiments, more advanced general FL methods should be compared. Moreover, adopting class-imbalanced loss in local training was proposed in several FL works [A, B] but missing. The idea of classifier calibration or using prototypes is also proposed before [CDE].

* Sharing the per-class gradients might weaken the privacy compared to vanilla FedAvg.


[A]  Federated visual classification with real-world data distribution, ECCV 2020
[B] On Bridging Generic and Personalized Federated Learning for Image Classification, ICLR 2022
[C] No Fear of Classifier Biases: Neural Collapse Inspired Federated Learning with Synthetic and Fixed Classifier, ICCV 2023
[D] No Fear of Heterogeneity: Classifier Calibration for Federated Learning with Non-IID Data, NeurIPS 2021
[E] Tackling data heterogeneity in federated learning with class prototypes, AAAI 2023

**Questions:**

I am not fully convinced by why the supplementary classifier can capture the global distribution precisely. Could the author provide more analysis on why and how precisely it can recover the global class distribution?

---

### Meta-Review · Area_Chair_QHc2 · 2023-12-04

**Metareview:**

This paper addresses a key challenge in FL when data samples have a long-tailed distribution. Decoupled-training that re-balances is proposed in the decentralized context, a method previously effective in centralized long-tailed learning.

The reviewers agree that the paper addresses an important problem, however, also had a couple of questions on the motivations, contributions and results of the paper. The authors did not answer in the rebuttal and the reviewer opinions did not change.

**Justification For Why Not Higher Score:**

The reviewers raised valid concerns, that were not addressed in the rebuttal.

**Justification For Why Not Lower Score:**

N/A

---

### Decision · Program_Chairs · 2024-01-16

Reject